# GRAPH FOURIER MMD FOR SIGNALS ON GRAPHS

## ABSTRACT

While numerous methods have been proposed for computing distances between probability distributions in Euclidean space, relatively little attention has been given to computing such distances for distributions on graphs. However, there has been a marked increase in data that either lies on graph (such as protein inter-action networks) or can be modeled as a graph (single cell data), particularly in the biomedical sciences. Thus, it becomes important to find ways to compare signals defined on such graphs. Here, we propose Graph Fourier MMD (GFMMD), a novel a distance between distributions and signals on graphs. GFMMD is defined via an optimal witness function that is both smooth on the graph and maximizes difference in expectation between the pair of distributions on the graph. We find an analytical solution to this optimization problem as well as an embedding of distributions that results from this method. We also prove several properties of this method including scale invariance and applicability to disconnected graphs. We showcase it on graph benchmark datasets as well on single cell RNA-sequencing data analysis. In the latter, we use the GFMMD-based gene embeddings to find meaningful gene clusters. We also propose a novel type of score for gene selection called *gene localization score* which helps select genes for cellular state space characterization.

## 1 INTRODUCTION

With the advent of high dimensional, high throughput data in fields ranging from biology, to finance, to physics, it becomes important to develop methods to perform high dimensional statistics in this sphere. In particular, the analysis of signals (or features in the data) and their pattern of spread through the landscape of data poses a challenge. If the signals act as functions on a low dimensional space like $\mathbb{R}^2$ or $\mathbb{R}^3$, it is possible to visualize them to gain a sense of the data. But what if these signals act on higher dimensional space like $\mathbb{R}^{9000}$? This could easily be the case in practice when a set of observations carries many variables, such as single-cell data. In order to handle data like this, one useful assumption has been that the data lies intrinsically in a lower-dimensional manifold $\mathcal{M}$, i.e., the Manifold hypothesis. This hypothesis has motivated low-dimensional embedding algorithms such as spectral clustering (Ng et al., 2001), tSNE (van der Maaten & Hinton, 2008), diffusion maps (Coifman & Lafon, 2006), and PHATE (Moon et al., 2019). In such algorithms, the data is first converted to an affinity graph, found by first computing distances between data points, and then affinities using a kernel function on the distances. This allows us to represent high dimensional data in a simpler form. Here, we use this representation to propose a distance between distributions or signals on such high dimensional data graphs called *Graph Fourier Maximum Mean Discrepancy* (GFMMD). Note, that GFMMD can also work on signals that naturally arise from graph or network structures, i.e., features of people in social interaction graphs.

Thus far while GNNs and other methods have focused on organizing and classifying nodes, there has been little focus on organizing the variables / signals themselves on these abstract spaces. For instance, in many measurements or sensors whose structure can be modelled as a graph, there is a need to understand the relationship between measured features. One example that arises in biology is that of single cell data. Here the cellular manifold can be modelled as a nearest neighbors graph, and each cell has measurements of thousands of genes, and there is a great deal of interest in understanding the relationships between genes and how and whether their expression is localized to parts of the cellular manifold.

Here we address the question of how to organize and compare signals on graphs in such a way that accounts for geometric structure on their underlying space. In particular, given a weighted graph $\mathcal{G} = (V, \mathcal{E}, w)$ and a set of functions $\{f_i\}_i$ on the vertices: $f_i : \mathcal{V} \rightarrow \mathbb{R}$, how can we structure and analyze these signals? We will first consider the case when $f_i$ is a probability mass function and extend the framework to arbitrary signals. This has a very natural applications to many modern datasets. To illustrate this, we focus on the application of embedding a set of genes on a graph of cells, as created from single cell RNA-sequencing data, and also in measuring whether the expression of a gene is localized (i.e., characteristic of a subpopulation of cells) or global like a house-keeping gene.

We present a new distance that belongs to the family of integral probability metrics (Sriperumbudur et al., 2012). Integral Probability Metrics (IPM) are distances between probability distributions that are characterized by a witness function that maximizes the discrepancy between distributions in expectation. The Maximum Mean Discrepancy (MMD) (Gretton et al., 2012) distances are a popular class of IPMs, they assume further structure in the space of witness functions, requiring that they come from a reproducing kernel Hilbert space.

Our notion of MMD, that we call *Graph Fourier MMD* (GFMMD), is a distance between signals on a data graph that is found by analytically solving for an optimal witness function. Furthermore, through the use of Chebyshev polynomials (Mason & Handscomb, 2002), GFMMD can be computed rapidly, and has a closed-form solution. We demonstrate its potential on toy datasets as well as single cell data, where we use it to identify gene modules.

Our main contributions are as follows: 1) We define Graph Fourier MMD as a distance between signals on arbitrary graphs, and prove that it is both an integrable probability metric and maximum mean discrepancy. 2) We derive an exact analytical solution for GFMMD which can be approximated in $\mathcal{O}(n(\log n + m^2))$ time to calculate all pairwise-distances between distributions, where $n$ is the number of vertices of the graph and $m$ is the number of signals. 3) We derive feature map for GFMMD that allows for efficient embeddings and dimensionality reduction. 4) We provide an efficient Chebyshev approximation method for computing GFMMD among a set of signals. 5) We showcase application of GFMMD to single cell RNA-sequencing data.

## 1.1 RELATED WORK

Spectral methods, such as (Coifman & Lafon, 2006; Belkin & Niyogi, 2003; Bronstein & Bronstein, 2010), define an embedding of the *nodes* of a graph using the eigendecomposition of a graph operator (Laplacian or diffusion operator). Similar to these methods, we use the graph's spectral properties to define an embedding of *signals* on the graph, and we show that this embedding preserve an MMD distance between signals.

The closest related work is that of Diffusion EMD Tong et al. (2021), which involves diffusion graph signals to different scales using a diffusion operator (similar to that of a diffusion map Coifman & Lafon (2006)) to create multiscale density estimates of the data. Then Diffusion EMD computes weighted $L^1$ distance between the multiscale density estimates of different signals. While this method is faster than most primal methods for EMD computation, it can be inaccurate unless the graph is significantly large.

Earlier methods that have been proposed for empirical estimations of high dimensional EMD include the Sinkhorn method Cuturi (2013), which involves Sinkhorn iterations (repeated normalization) of a joint probability distribution to converge at a distribution that describes a valid transport plan, i.e. whose marginals agree with the two empirical distributions. The authors of (Solomon et al., 2015; Huguet et al., 2022) extent the Sinkhorn algorithm to graphs with a heat-geodesic ground distance. Their algorithm can be computed efficiently for two signals on a sparse graphs, but does not provide an embedding of signals.

In (Le et al., 2022; 2019; Essid & Solomon, 2018), the authors consider the EMD between distributions defined on a *distance* graph, that is the edge weights define the cost of moving mass from one node to another. The authors in (Le et al., 2022; 2019) provide a closed-form solution that relies on a graph shortest path distance. In this setting, there is no sparse approximation to diffusion distances in terms of graph shortest path. We consider a different problem where the edges of the graph are affinities.

Among methods for MMD, the most common method has been a sampling based method that also forms a 2-sample Kernel test based on defining a kernel between empirical observations Gretton et al. (2012). Note that semantically this takes distances between point clouds themselves by modeling them as a data graph with vertices as points. We define a method of taking signals which generalizes to an arbitrary graph, on a point cloud or otherwise, and demonstrate its effectiveness both when the graph lies in a metric space and when adjacencies are binary. We compare our method GFMMD to all three of these methods.

## 2 Preliminaries

**Integral probability metrics** IPMs (Müller, 1997; Sriperumbudur et al., 2012) constitute a family of distances between probability distributions. They are often used when dealing with empirical samples (datasets) sampled from a continuous space. In contrast, the alternative class of $\phi$-divergences (such as KL-divergence) is often less useful as a measure between empirical samples with poor behavior when the domains do not overlap. In contrast to $\phi$-divergences, integral probability metrics are defined over a metric space, this allows for a reasonable distance between distributions with non-overlapping support.

**Definition 1.** *Given a metric space $(\mathcal{X}, d)$, a family $\mathcal{F}$ of measurable, bounded functions on $\mathcal{X}$, and two measures $P$ and $Q$ on $\mathcal{X}$, the IPM between $P$ and $Q$ is defined as*

$$\gamma_{\mathcal{F}}(P, Q) \triangleq \sup_{f \in \mathcal{F}} \mathbb{E}_P(f) - \mathbb{E}_Q(f).$$

Here, $\mathcal{F}$ is a family of "witness function" since it emphasizes the differences between $P$ and $Q$, choosing a certain $\mathcal{F}$ determines the IPMs.

**Maximum Mean Discrepancy** If $\mathcal{H}$ is a Reproducing Kernel Hilbert Space (RKHS) of functions on $\mathcal{X}$ (equipped with norm $\|(\cdot)\|_{\mathcal{H}}$), then the IPM corresponding to $\mathcal{F} = \{f : \|f\|_{\mathcal{H}} \leq 1\}$ is a *kernel Maximum Mean Discrepancy* (Gretton et al., 2012). Numerous distances between distributions are IPMs, given a suitable choice of $\mathcal{F}$. For example, the Wasserstein distance is an IPM where $\mathcal{F}$ corresponds to the family of Lipschitz functions.

Computing MMDs in practice could be difficult. Algorithms like Kernel MMD rely on the fact that kernels induce such a RKHS, and vice-versa. It can be shown that, for a given positive definite kernel $k : \mathcal{X} \times \mathcal{X} \to \mathbb{R}$, there exists a RKHS $\mathcal{H}$ in which, $\forall f \in \mathcal{H}$, $f(x) = \langle f, k(x, \cdot) \rangle_{\mathcal{H}}$. So in particular, $\langle k(y, \cdot), k(x, \cdot) \rangle_{\mathcal{H}} = k(x, y)$.

**Definition 2.** *Let $k(\cdot, \cdot)$ be a positive definite kernel and $\mathcal{H}$ is the Hilbert space in which $\langle f, k(x, \cdot) \rangle_{\mathcal{H}} = f(x)$ for all $f \in H$. The mean embedding of a probability distribution $P$ is defined as $\mathbb{E}_{X \sim P}[k(X, \cdot)]$, and the Maximum Mean Discrepancy between $P$ and $Q$ is defined to be $\mathrm{MMD}(P, Q) = \sup_{\|f\|_{\mathcal{H}} \leq 1} \mathbb{E}_P(f) - \mathbb{E}_Q(f)$.*

As a consequence of the Cauchy-Schwarz inequality, it can be shown that $\mathrm{MMD}(P, Q) = \|\mu_P - \mu_Q\|_{\mathcal{H}}$. Kernel MMD (Gretton et al., 2012) works by estimating the norm in the RKHS with the empirical mean embeddings defined by the kernel $k(\cdot, \cdot)$

$$\|\mu_P - \mu_Q\|_{\mathcal{H}}^2 = \mathbb{E}_{X, X' \sim P} k(X, X') + \mathbb{E}_{Y, Y' \sim Q} k(Y, Y') - 2 \mathbb{E}_{X \sim P, Y \sim Q} k(X, Y).$$

**Example 2.1.** *If $X \sim P$ and $Y \sim Q$ are two point clouds of size $m$ and $n$, we can approximate the distance $MMD(P, Q)$ by approximating the expectations with empirical means*

$$\approx \frac{1}{m(m-1)} \sum_{i=1}^{m} \sum_{j=1, j \neq i}^{m} k(x_i, x_j) + \frac{1}{n(n-1)} \sum_{i=1}^{n} \sum_{j=1, j \neq i}^{n} k(y_i, y_j) - 2 \frac{1}{mn} \sum_{i=1}^{m} \sum_{j=1}^{n} k(x_i, y_j).$$

*Note that, in practice, this quantity is often computed by random sampling with replacement from the point clouds $X$ and $Y$.*

**Wasserstein Distance**    The Earth Mover's Distance (EMD), also known as the 1-Wasserstein distance, is a distance between probability distributions designed to measure the least amount of "work" it takes to move mass from one distribution to another. Formally, we are given two distributions $P$ and $Q$ on a measure space $(\Omega, \mathcal{F}, \mu)$ and a distance $d : \mathcal{X} \times \mathcal{X} \to \mathbb{R}$. Most commonly, $\Omega$ might be a Riemann manifold, $\mathbb{R}^d$, or in our case, a finite graph. We define the space of couplings of $P$ and $Q$, denoted $\Pi(P, Q)$ to be the set of joint probability distributions whose marginals are equal to $P$ and $Q$.

**Definition 3.** *The 1-Wasserstein Distance between $P$ and $Q$ is defined to be:*

$$W(P, Q) \triangleq \min_{\pi \in \Pi(P,Q)} \int_{\mathcal{X} \times \mathcal{X}} d(x, y) \pi(x, y).$$

The supremizing joint distribution $\pi$ would then be called the *optimal transport plan*. In the case that $\Omega$ is finite (say of size $n$), $\Pi(P, Q)$ could be thought of as the set of $n \times n$ matrices $\pi$ for which $\pi \mathbf{1} = P, \mathbf{1}^T \pi = Q$. Then we could represent distances in a $n \times n$ matrix $D$, and the EMD is given by $\min_{\pi} \pi \cdot D$. Typical solutions to EMD in its *primal form* are found using linear programming. A well known theorem in the study of optimal transport is the Kantorovich-Rubinstein Duality, which provides an equivalent expression for the Wasserstein Distance based on the dual optimization. Letting $\|f\|_{\leq 1}$ symbolize the condition that $f$ obeys a 1-Lipschitz constraint: in other words, that for all $x, y \in \mathcal{X}$, $|f(x) - f(y)| \leq d(x, y)$.

**Theorem 1.** *(Kantorovich-Rubinstein) The EMD is an IPM with $\mathcal{F}$ the space of 1-Lipschitz functions*

$$W(P, Q) = \sup_{\|f\|_{\leq 1}} \mathbb{E}_P(f) - \mathbb{E}_Q(f).$$

We refer to Dudley (2018) for a proof of the previous theorem. Intuitively, we can think of suitable functions $f$ as being varying slowly over $\mathcal{X}$. The 1-Lipschitz constraint prevents witness functions from behaving too erratically over the space. Then calculating the Wasserstein distance is equivalent to solving the above linear program. In the discrete case, we are trying to maximize $\langle f, P \rangle - \langle f, Q \rangle$ subject to $|f(a) - f(b)| \leq d(a, b)$ for all $a, b \in \mathcal{V}$.

**The Graph Laplacian**    For a weighted graph $\mathcal{G} = (\mathcal{V}, \mathcal{E}, w)$ on $n$ vertices, we have a number of associated matrices. The first of which is an adjacency / affinity matrix $\mathbf{A}$ for which, given vertices $a$ and $b$, $\mathbf{A}(a, b) = w(a, b)$; for our purposes, we assume $w(a, b) \geq 0$. In the case when $\mathcal{V}$ belongs to a metric space $(\mathcal{X}, d)$, we have an associated distance matrix $M$ for which $M(a, b) = d(a, b)$ for all $a, b \in \mathcal{V}$. Oftentimes, the affinity matrix $\mathbf{A}$ is generated by a nonlinear kernel function $k(\cdot)$ so that $\mathbf{A}(a, b) = k(M(a, b))$. For our purposes, if $\mathbf{A}$ is generated in this way, we will call $\mathcal{G}$ a *affinity graph*. There is also a diagonal degree matrix for which $\mathbf{D}(a, a) = \sum_{b \in \mathcal{V}} w(a, b)$. Finally, we define the combinatorial Laplacian $\mathbf{L} = \mathbf{D} - \mathbf{A}$. It can be shown that for any function on the vertices $f$, $f^T \mathbf{L} f = \sum_{(a,b) \in \mathcal{E}} w(a, b)(f(a) - f(b))^2$. From this identity, it's clear that $\mathbf{L}$ is positive semi-definite, with spectrum $\{\lambda_i, \psi_i\}_{i=0}^{n-1}$ (such that $\lambda_i$ is nondecreasing in $i$). In fact, the dimension of $\mathbf{L}$'s kernel is equal to the number of connected components of $\mathcal{G}$. In particular, $\mathbf{L}\mathbf{1} = 0$, where $\mathbf{1}$ is the all 1's vector. If $\mathcal{G}$ is fully connected, then $\mathbf{1}$ spans the kernel of $\mathbf{L}$. Since $\mathbf{L}$ is positive semi-definite, it is diagonalizable with an orthonormal set of eigenvectors, so we may write $\mathbf{L} = \Psi \Lambda \Psi^T$ where $\Lambda = \text{diag}(\lambda_0 \, \lambda_1 \ldots \lambda_{n-1})$ and $\Psi = (\psi_0 \, \psi_1 \ldots \psi_{n-1})$. Then the *Graph Fourier Transform* of a function $f : \mathcal{V} \to \mathbb{R}$ is defined as $\hat{f} = \Psi^T f$. Similarly, for any function $h : \{\lambda_i\}_i \to \mathbb{R}$, we can define the graph filter $H : f \mapsto \sum_{i=0}^{n-1} h(\lambda_i) \hat{f}(i) \psi_i = \Psi h(\Lambda) \Psi^T f$. As a matter of notation, the $\mathbf{L}^-$ refers to the Moore-Penrose inverse of $\mathbf{L}$ and $\mathbf{L}^{-1/2}$ is defined as $\Psi \Lambda^{-1/2} \Psi^T$, where $\Lambda^{-1/2}$ is obtained by taking the elementwise reciprocal square roots of positive entries of $\Lambda$, and leaving the zeros as is.

## 3    METHODS

In the following, with present definitions and theorems for arbitrary probability distributions $P$ and $Q$, but our theory can be extended to any real bounded signals, the expectations would be replaced by the integrals w.r.t. a finite (signed) measure.

### 3.1 Graph Fourier MMD as an Optimization

For Graph Fourier MMD in a metric space, we are given a finite affinity graph $\mathcal{G} = (\mathcal{V}, \mathcal{E}, d)$ on a metric space $(\mathcal{X}, d)$. We are also given two probability distributions $P, Q : \mathcal{V} \to \mathbb{R}$ (where each node comes with a given probability). If we were to take the 1-Wasserstein distance between $P$ and $Q$, the dual form would yield:

$$W(P, Q) = \max_f \langle f, P - Q \rangle \qquad \text{subject to} \qquad \frac{1}{d(a,b)^2}(f(a) - f(b))^2 \leq 1 \quad \forall a, b \in \mathcal{V}$$

Now, we will augment the above constraint. Instead, we impose the new relaxed constraint that $\sum_{(a,b) \in \mathcal{E}} \frac{1}{d(a,b)^2}(f(a) - f(b))^2 \leq T$ for some threshold $T$. Note that if $w(a, b) = 1/d(a, b)^2$, then this condition is equivalent to $f^T \mathbf{L} f \leq T$. The constraint $f^T \mathbf{L} f \leq T$ is valid for any way of calculating the affinity and result in a closed-form solution. It as a similar interpretation than the 1-Lipschitz as it quantifies the smoothness of a signal on a graph with respect to the affinities. For example, if we assume that our data lies on a manifold, we can choose a different level of affinity, such as $w(a, b) = \exp(-d(a, b)^2/2\sigma^2)$ to penalize transport over larger global distances. Thus, we define our distance between $P$ and $Q$ only in terms of $\mathbf{L}$. In the following, we only assume that the affinities are nonnegative.

**Definition 4.** *Let $\mathcal{G} = (\mathcal{V}, \mathcal{E}, w)$ be a finite graph with Laplacian $\mathbf{L}$ and $P, Q$ be two bounded probability distributions on $\mathcal{V}$. For $T \in \mathbb{R}^+$, the Graph Fourier MMD between $P$ and $Q$ is*

$$\textit{GFMMD}(P, Q) \triangleq \max_{f : f^T \mathbf{L} f \leq T} \mathbb{E}_P(f) - \mathbb{E}_Q(f).$$

Note that this definition holds for any construction of a positive semi-definite Laplacian matrix $\mathbf{L}$ and chosen $T$.

**Theorem 2.** *Let $\mathcal{G}, \mathbf{L}, T$ be as defined in Definition 4 and let $\mathcal{G}$ be fully connected. Then, for any two bounded probability distributions $P$ and $Q$ defined on $\mathcal{V}$, $\textit{GFMMD}(P, Q) = \sqrt{T}\|\mathbf{L}^{-\frac{1}{2}}(P - Q)\|$, where $\|\cdot\|$ is the $\ell_2$ norm in $\mathbb{R}^n$.*

*Proof.* $P$ and $Q$ can be viewed as vectors indexed over $\mathcal{V}$ so that, for any function $f : \mathcal{V} \to \mathbb{R}, \mathbb{E}_P(f) - \mathbb{E}_Q(f) = \langle P - Q, f \rangle$. Additionally, note that for any function $f$, we can write $f = f_1 + f_2$, where $f_1$ is orthogonal to the $\mathbf{1}$ vector (so $f_1 \in C(\mathbf{L})$) and $f_2 \in \ker(\mathbf{L})$ the kernel of $\mathbf{L}$. Since $\ker(\mathbf{L})$ is $\mathbf{1}$, we have $f_2 = c\mathbf{1}$ for some $c \in \mathbb{R}$. Then, $\mathbb{E}_P(f) - \mathbb{E}_Q(f) = \mathbb{E}_P(f_1) - \mathbb{E}_Q(f_1) + \mathbb{E}_P(f_2) - \mathbb{E}_Q(f_1) = \mathbb{E}_P(f_1) - \mathbb{E}_Q(f_1) + \mathbb{E}_P(c\mathbf{1}) + \mathbb{E}_Q(c\mathbf{1}) = \mathbb{E}_P(f_1) - \mathbb{E}_Q(f_1) + c\mathbf{1} - c\mathbf{1} = \mathbb{E}_P(f_1) - \mathbb{E}_Q(f_1)$. Thus, for any such $f$, it can be assumed, without loss of generality, that $f \perp \mathbf{1}$. If $\mathcal{G}$ is disconnected, we can still assume that $f \perp \ker(\mathbf{L})$, but this is more technical and will be omitted for now (see Theorem A.1.1). In particular, it can be assumed that $\mathbf{L}\mathbf{L}^- f = f$ where $\mathbf{L}^-$ is the pseudo-inverse of $\mathbf{L}$. Note $f^T \mathbf{L} f \leq T$ if and only if $|\mathbf{L}^{\frac{1}{2}} f| \leq \sqrt{T}$. Thus, we can rewrite the optimization as $\max\langle P - Q, \sqrt{T}\mathbf{L}^{-\frac{1}{2}} \cdot \frac{1}{\sqrt{T}} \cdot \mathbf{L}^{\frac{1}{2}} f \rangle$ where the maximum is over $\{f : |\frac{1}{\sqrt{T}}\mathbf{L}^{\frac{1}{2}} f| \leq 1\}$. Doing a change of variables of $\mathbf{L}^{\frac{1}{2}} f$ to $x$, then from the Cauchy Schwarz Inequality, we have $\text{GFMMD}(P, Q) = \sqrt{T} \sup_{\|x\| \leq 1} \langle \mathbf{L}^{-\frac{1}{2}}(P - Q), x \rangle = \sqrt{T}\|\mathbf{L}^{-\frac{1}{2}}(P - Q)\|$, as desired. $\square$

Note that the assumption that $\mathcal{G}$ is connected was key to this proof. These concerns are addressed in Theorems A.1.1 and Corollary A.1.2. Since $P - Q \notin C(\mathbf{L})$ if and only if $\langle (P-Q), \mathbb{I}\{C_k\} \rangle = 0$ for all connected components $k$, this condition can be easily checked in practice. GFMMD possesses a set of convenient properties. Namely, we have a representation in terms of an explicit feature map $\mathbf{L}^{-\frac{1}{2}}$. So to compute pairwise distances, it is sufficient to apply the feature map and then take Euclidean distances. Also, the distance value is a bona fide distance (particularly an MMD), and it is also preserved under certain graph manipulations. These are the results of Lemmas 3 and 4.

**Lemma 3.** *(i) GFMMD$(\cdot, \cdot)$ defines a valid distance on the probability distributions acting on $\mathcal{V}$. Furthermoe, (ii) GFMMD$_{\mathcal{G}}(P, Q)$ is a Maximum Mean Discrepancy with explicit feature map $\sqrt{T}\mathbf{L}^{-\frac{1}{2}}$.*

*Proof.* For (i), note that $\mathbf{L}^{-1/2}(P - Q)$ is linear. By the usual nonnegativity of lengths, GFMMD$(P, Q) = \sqrt{T}\|\mathbf{L}^{-\frac{1}{2}} P - \mathbf{L}^{-\frac{1}{2}} Q\| \geq 0$, so GFMMD$(\cdot, \cdot)$ is nonnegative. Furthermore,

note that $\text{GFMMD}(P, Q) = 0$ if and only if $\mathbf{L}^{-\frac{1}{2}}P = \mathbf{L}^{-\frac{1}{2}}Q$. But since $P$ and $Q$ sum to 1, $\mathbf{L}^{-\frac{1}{2}}$ as injectively on the set of functions orthogonal to its kernel, so $P = Q$. Thus, $\text{GFMMD}(P, Q) \geq 0$, with equality if and only if $P = Q$. Finally, the triangle inequality holds, since for arbitrary probability densities $P, Q, R$, $\text{GFMMD}(P, Q) = \sqrt{T}\|\mathbf{L}^{-\frac{1}{2}}P - \mathbf{L}^{-\frac{1}{2}}Q\| \leq \sqrt{T}\|\mathbf{L}^{-\frac{1}{2}}P - \mathbf{L}^{-\frac{1}{2}}R\| + \sqrt{T}\|\mathbf{L}^{-\frac{1}{2}}Q - \mathbf{L}^{-\frac{1}{2}}R\| = \text{GFMMD}(P, R) + \text{GFMMD}(Q, R)$ follows from the usual triangle inequality in $\ell_2$. Thus, GFMMD is a valid distance acting on probability distributions. For (ii), by definition, an MMD $\gamma$ between $P$ and $Q$ takes the form $\gamma(P, Q) = \sup_{\|f\|_{\mathcal{H}} \leq 1} \mathbb{E}_P(f) - \mathbb{E}_Q(f)$, where $\mathcal{H}$ is some Hilbert Space and $\{f : \|f\|_{\mathcal{H}} \leq 1\}$ corresponds to the unit ball. If we define a Hilbert space on $\ell_2$ with $\langle x, y \rangle_{\mathcal{H}} = \frac{1}{T}x\mathbf{L}^- y$, it follows that $\|f\|_{\mathcal{H}} \leq 1$ corresponds to $f^T \mathbf{L} f \leq T$. Thus, $\text{GFMMD}(\cdot, \cdot)$ possesses the form of a valid MMD. Therefore, this distance is also an IPM. $\square$

Note that if we choose our parameter $T$ in such a way that it is proportional to the total degree of the graph, we may rescale our affinities however we like without affecting the underlying distance. This is the claim of Lemma 4.

**Lemma 4.** *If $\mathcal{G} = (\mathcal{V}, \mathcal{E}, w)$ and $\mathcal{G}' = (\mathcal{V}, \mathcal{E}, cw)$ are finite, fully connected graphs with $c \in \mathbb{R}$ and $T$ is chosen to be proportional to the total degree of the graph, then $GFMMD_{\mathcal{G}}(P, Q) = GFMMD_{\mathcal{G}'}(P, Q)$. In other words, Graph Fourier MMD is scale invariant.*

*Proof.* This follows from the fact that a matrix scales inversely with its pseudo-inverse. If $T$ is proportional to the total graph degree $T \propto \sum_a \mathbf{D}(a, a)$, then we define $T = k \cdot \sum_a \mathbf{D}(a, a)$. In particular, if $\mathbf{L}, \mathbf{L}'$ are the Laplacians of $\mathcal{G}$ and $\mathcal{G}'$, then all of the following hold: $\mathbf{L}' = c\mathbf{L}, T = k \operatorname{Tr} \mathbf{L}, T' = k \operatorname{Tr} \mathbf{L}' = k \operatorname{Tr} c\mathbf{L} = cT$. Also note $(\mathbf{L}')^-(c\mathbf{L})^- = \frac{1}{c}\mathbf{L}^-$, so $\sqrt{T'}\|(\mathbf{L}')^-(P - Q)\| = \sqrt{cT}\|\frac{1}{c}\mathbf{L}^-(P - Q)\| = \sqrt{T}\|(\mathbf{L})^-(P - Q)\|$. And we conclude $\text{GFMMD}_{\mathcal{G}'}(P, Q) = \text{GFMMD}_{\mathcal{G}}(P, Q)$ $\square$

A nice corollary of Lemma 4 is that if our affinities satisfy $w(a, b)/d(a, b) = g(c)$ for some function $g$ of $c$, then GFMMD is invariant under rescaling of distances as well. For instance, by doubling all points in $\mathcal{V}$, the GFMMD between two distributions remains the same. Moreover, if we were to choose to encode each vertex $a$ of $\mathcal{G}$ as the Kronecker delta $\delta_a$, then the corresponding embeddings for each vertex would be familiar. In fact, the best $k$-dimensional representation (by multidimensional scaling) of the vertices will coincide almost exactly with Hall's Spectral Graph Drawing (Hall, 1970), which uses the first $k$ nontrivial eigenvectors to represent vertices using coordinates in $\mathbb{R}^k$. This is made formal by Theorem 5, which proof is presented in the Appendix.

**Theorem 5.** *If $X = \{\delta_i\}_{i \in \mathcal{V}}$ is a family of Kronecker-delta functions centered at each vertex of $\mathcal{G}$, then the $k$-dimensional embedding which best preserves the distances between signals in $X$ is equivalent up to rescaling to Hall's Spectral Graph Drawing of the Graph $\mathcal{G}$ in $k$-dimensions.*

## 3.2 COMPUTATIONAL COMPLEXITY AND SPEEDUP

Computation of exact GFMMD is $O(n^3)$ as it requires computing the pseudo-inverse of $\mathbf{L}$. The space complexity is $O(n^2)$ for since it requires representation of the full distance matrix. However, with some minor modifications we can greatly reduce the time and memory requirements in the particular case where the graph $\mathcal{G}$ is sparse i.e. $|E| = O(n \log n)$. In such cases, we present an $O(n \log n)$ algorithm for the computation of GFMMD which is substantially faster than naive implementations based on a Chebyshev polynomial approximation of the filter in Algorithm 1 as well as a KNN kernel. The steps for an arbitrary graph are the same, but with $\mathbf{W}$ provided.

---

**Algorithm 1** GFMMD in Metric Space

---

**Input:** A set of $n$ points $X \subseteq \mathbb{R}^d$, $m$ probability distributions $f_i : X \to \mathbb{R}$ in an $n \times m$ matrix $F$, and a kernel function $k : X \times X \to \mathbb{R}$
**Output:** An $m \times n$ embedding matrix $\boldsymbol{E}$ in which $\|\boldsymbol{E}_i - \boldsymbol{E}_j\| = \text{GFMMD}(f_i, f_j)$. $M$ and a distance matrix in which $\boldsymbol{M}_{ij} = \text{GFMMD}(f_i, f_j)$

Create a thresholded K-Nearest Neighbor graph $\mathcal{G}$ over $X$ with $\mathcal{O}(n \log n)$ edges
$\boldsymbol{W}_{ij} \leftarrow k(X_i, X_j)$ for all $(i, j) \in \mathcal{E}$
$\mathbf{L} \leftarrow \mathbf{D} - \mathbf{W}$, where $\mathbf{D}$ is the matrix whose diagonal is the row sums of $\mathbf{W}$.
$\boldsymbol{E}_i \leftarrow \mathbf{L}^{-\frac{1}{2}} f_i$ either by Chebyshev approximation of the filter $h(\lambda) = \lambda^{-\frac{1}{2}}$
$\boldsymbol{M}_{ij} \leftarrow \|\boldsymbol{E}_i - \boldsymbol{E}_j\|$ for all $i, j \in [n]$
return $\boldsymbol{M}, \boldsymbol{E}$

---

This gives a total fine grained time complexity of $\mathcal{O}((k_1+t)n \log n + k_2 nm \log m)$ and a space complexity of $O(n \log n + mn)$ space. Here, $t$ is the order of the Chebyshev polynomial, $k_1$ is the threshold for number of nearest-neighbors in constructing $\mathcal{G}$, and $k_2$ is the number of nearest-distributions we'd like to calculate. More simply, for fixed Chebyshev order, and number of neighbors, the time to estimate distances between all distributions is $\mathcal{O}(n \log n + nm^2)$.

## 4 EMPIRICAL RESULTS & APPLICATIONS

In our experiments, signals are always nonnegative and normalized to be interpreted as probability distributions. However, while Graph Fourier MMD is developed in the case in which $P$ and $Q$ are regarded as signals representing probability distributions, we may take distances between any two bounded signals $P$ and $Q$ on a shared graph via $\text{GFMMD}(P, Q) = \|\mathbf{L}^{-1/2}(P - Q)\|^2$.

### 4.1 IDENTIFYING DISTRIBUTIONS ON THE SWISS ROLL

In this experiment, we generate random point clouds centered at points on the swiss roll. More specifically, we sample $n = 100$ points on the swiss roll $x_1, x_2, \ldots, x_n$ and rotate the coordinates into 10-dimensional space. Then, around each of these points, we generate a point cloud $d_i$ of size $m = 100$ points from a multivariate normal distribution centered at $x_i$. The result is $nm$ points in $\mathbb{R}^{10}$. For each $i, j \in [n]$, we have a known geodesic distance between $x_i$ and $x_j$. Across the different measures, we can see how well the distance between the point clouds $d_i$ and $d_j$ compares to the geodesic distance between their corresponding centers $x_i$ and $x_j$. We generate a common data graph with $nm$ vertices and consider a family of $n$ probability distributions $p_i(x) = \frac{1}{m}\mathbf{1}\{x \in d_i\}$ We compare the distributions using: 1) computation of earth Mover's Distance between point clouds in ambient space, 2) Sinkhorn algorithm (Cuturi, 2013), 3) Diffusion EMD (Tong et al., 2021), 4) Kernel MMD (Gretton et al., 2012) between all pairs $p_i, p_j$ via random sampling (20 points from each distribution with replacement), 5) Graph Fourier MMD between $p_i, p_j$, using both the exact calculation and approximation via Chebyshev polynomials. Results are shown in Table 1.

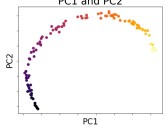 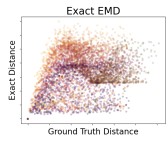 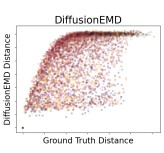 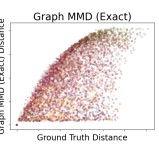 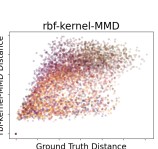

Figure 1: Left: first two PCs of the embeddings $E$ from Algorithm 1., colored by the coordinate of the corresponding center along the curved direction of the swiss roll. Right: Geodesic distance between centers vs. corresponding distance between distributions

By plotting true geodesic distance between centers against distance by each of these methods, we can assess the corresponding method accuracy. On the left of Figure 1 is EMD, where the oscillatory pattern illustrates its ineffectiveness at calculating distances between distributions on graphs, since Euclidean distance between points on the swiss roll has periodic behavior in curvature. Diffusion EMD and Kernel MMD are effective at taking distances between points initially, but fail to discern

| Method | Spearman-$\rho$ | 10-NN time (s) | All-pairs time(s) |
|---|---|---|---|
| DiffusionEMD | $0.584 \pm 0.017$ | $2.171 \pm 0.265$ | $3.341 \pm 0.333$ |
| Exact | $0.253 \pm 0.022$ | $26.881 \pm 1.104$ | $26.881 \pm 1.104$ |
| Sinkhorn | $0.250 \pm 0.022$ | $54.346 \pm 17.576$ | $54.346 \pm 17.576$ |
| rbf-kernel-MMD | $0.509 \pm 0.021$ | $5.016 \pm 0.237$ | $5.016 \pm 0.237$ |
| Graph MMD (Exact) | $\mathbf{0.613 \pm 0.019}$ | $139.453 \pm 16.790$ | $139.468 \pm 16.794$ |
| Graph MMD (Chebyshev, 8) | $0.606 \pm 0.024$ | $\mathbf{0.619 \pm 0.057}$ | $\mathbf{0.641 \pm 0.056}$ |
| Graph MMD (Chebyshev, 64) | $0.593 \pm 0.021$ | $1.155 \pm 0.035$ | $1.163 \pm 0.035$ |
| Graph MMD (Chebyshev, 512) | $0.612 \pm 0.018$ | $6.249 \pm 2.896$ | $6.258 \pm 2.895$ |
| Graph MMD (Chebyshev, 4096) | $0.612 \pm 0.018$ | $48.138 \pm 1.184$ | $48.159 \pm 1.182$ |

Table 1: Comparison of runtime and Spearman-$\rho$ correlation to ground truth manifold distances between distributions with mean $\pm$ standard deviation over 10 seeds for 100 distributions of 100 points each on a swiss roll manifold. The exact Graph MMD is most performant but requires a eigen-decomposition. The Chebyshev approximated Graph MMD (Chebyshev, $t$) is extremely fast and almost as performant at even lower orders $t$.

between higher and higher distances. Graph Fourier MMD, on the other hand, has a far more clear linear correlation, which levels off much slower. If instead of taking the distances in the embedded space between distributions, we could take the embeddings and project them to a low dimensional space by running PCA on the matrix $\sqrt{T}\mathbf{L}^{-1/2}X$, where $X = [d_1 \quad d_2 \ldots \quad d_n]$. We would therefore obtain the best lower dimensional linear embedding in which distances between distributions approximates their distance according to Graph Fourier MMD. By taking the first two PCs of the embedding and coloring by the curvature coordinate of corresponding distribution center, we find that we are highly effective at capturing geometry of the underlying manifold.

## 4.2 SINGLE CELL ANALYSIS WITH GFMMD

To demonstrate the utility of Graph Fourier MMD for biological analysis, we leverage publicly available single-cell RNA sequencing dataset of CD8-positive T cells (Zheng et al., 2017). CD8-positive T cells are adaptive immune cells known to be critical for mediating immune response in infection, cancer, and other diseases. This dataset consists of 9,167 cells with 1,991 highly variable genes on each cell. Then we apply Algorithm 1 with the adaptive Gaussian Kernel (Moon et al., 2019) between datapoints, to compute GFMMD between genes, resulting in an $1991 \times 1991$ matrix of distances. Here we viewed each gene as a distribution nearest neighbor cell graph. We also obtain $E$ which is an $1991 \times 9167$ matrix containing an explicit feature map or embedding of the genes. In Figure 2A, we visualize the gene embedding using both PCA and PHATE (Moon et al., 2019). We find that clusters $0 - 9$ in from the gene embedding, show characteristic expression on the cellular embedding in Figure 2B. In other words, the subplots in Figure 2B represent a PHATE map of the cells in this dataset, and when we highlight the expression of gene clusters on the cells we see that these clusters have localized expression on the cellular manifold.

To interpret these gene clusters for biological significance, we analyzed the gene set enrichment of clusters 6 and 7 with Enrichr (Chen et al., 2013), which show high expression in opposite ends of the cellular manifold (see Figure 2. Enrichr shows that cluster 7 has strong enrichment for signatures of a naive T cell becoming activatied with mitosis and T cell activation signatures being significant. On the other hand, cluster 6 shows strong enrichment for an effector CD8 T cell, with signatures of cytotoxic activity and inflammatory signaling (interferon gamma). Thus, these genes can be used to characterize the cellular manifold as following a trajectory from naive to effector CD8 T cells. We compare these to gene clusters derived from DiffusionEMD, as well as to a more standard method of gene selection in biology: differential expression of genes in different areas of cellular state space based on a Wilcoxon rank sum test between the two manually curated cell clusters from Zheng et al. (2017). The gene clusters 4 and 8 from DiffusionEMD that were most enriched on the opposite ends of the manifold consisted of 6 genes and 11 genes, which resulted in no enrichment for the above signatures. These genes upregulated based on the Wilcoxon rank sum test give a much less clear picture of the cellular state space, with the same annotations scoring much lower.

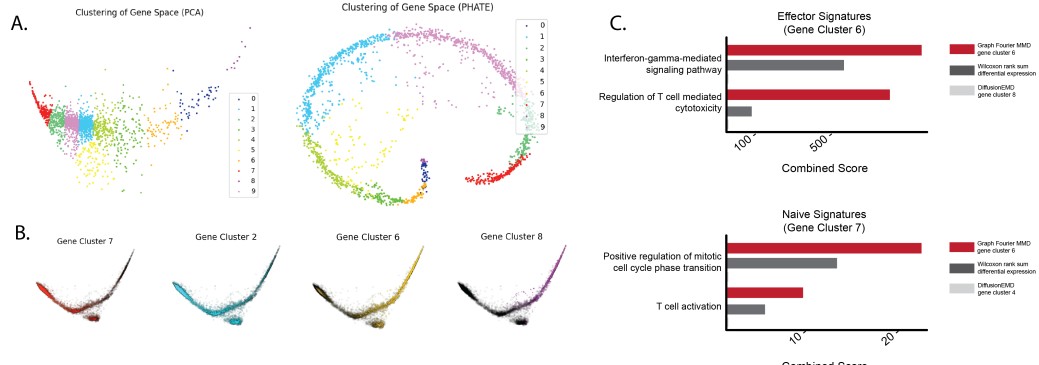

Figure 2: A. Embeddings of genes of the dataset from Zheng et al. (2017) computed by the GFMMD Algorithm, visualized with PCA and PHATE (Moon et al., 2019), colored by results of K-means clustering. B. Embeddings of cells from Zheng et al. (2017) visualized with PHATE. Each plot is colored by the average expression of genes in the marked cluster over cells. C. Comparison of enrichment scores from Enrichr (Chen et al., 2013) on T-cell relevant annotations, between GFMMD gene sets (clusters 7,6) expression-based gene sets.

**Local Genes** A novel type of analysis enabled by GFMMD is a search for *localized* signals. Often, researchers in the single cell field search highly variable genes, but we posit that genes that have localized expression on cellular manifolds can be used to characterize salient cellular subtypes. We propose a measure of gene locality by GFMMD between the gene signal and a uniform distribution on the graph.

**Definition 5.** *The* gene localization score, *s of gene p, described by distribution $P$ on the cellular data graph as:* $s(p) = GFMMD(P, U)$. *Here,* $U = \frac{1}{n}\mathbf{1}$ *on the same vertex set.*

**Theorem 6.** *The localization score s can be computed as:* $s(p) = \mathrm{GFMMD}(P, U) = \|\mathbf{L}^{-\frac{1}{2}}P\|$.

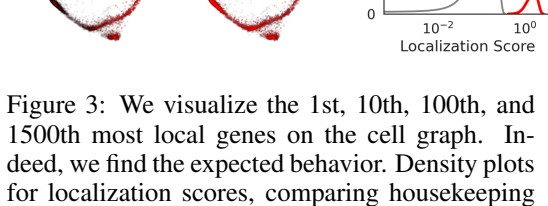

Figure 3: We visualize the 1st, 10th, 100th, and 1500th most local genes on the cell graph. Indeed, we find the expected behavior. Density plots for localization scores, comparing housekeeping genes and naive CD8+ T cell signature. The naive gene signatures are given by the red curve and Housekeeping gene signatures by the gray.

*Proof.* Since $\mathbf{1}$ is in the kernel of $\mathbf{L}$ (and thus $\mathbf{L}^-$), we have $\mathrm{GFMMD}(P, U) = \|\mathbf{L}^{-\frac{1}{2}}P - \mathbf{L}^{-\frac{1}{2}}\mathbf{1}\| = \|\mathbf{L}^{-\frac{1}{2}}P\|$. □

Based on this score, in Figure 3, we visualize first most local gene, 10th most local gene, and 20th most local. Here, we compare localization scores between housekeeping genes and the gene signature for naive CD8+ T cells. Housekeeping genes are expressed highly in many systems, but are not known to have a function that contributes strongly to cell-cell variation for T cells Eisenberg & Levanon (2003); Wang et al. (2021); de Jonge et al. (2007). By contrast, cells enriched for the naive CD8+ T cell signature are a subset of T cells along the T cell differentiation axis. We show that the localization score is an order of magnitude higher for the naive gene signature versus the housekeeping signature Figure 3, validating our intuition about localized genes.

## 5 CONCLUSION

In this paper we have introduced Graph Fourier MMD, a framework for taking distances between signals on graphs and generating embeddings in which these distances hold. We have shown its intuitive performance in both the Riemannian and abstract graphical setting for known distributions, as well as its advantage in speed, and ability to capture global properties of the underlying data manifold compared to alternative methods like Earth Mover's Distance and Diffusion EMD. Its rapidity makes it particularly useful for high dimensional datasets, such as single cell data, where we have showed its ability to capture the natural trajectories of gene expression.

## 6 REPRODUCIBILITY

In the experiments, the main libraries which were used were pygsp to generate, store, and manipulate graphs. This includes key functions such as filter, which we use for both the exact and Chebyshev approximated version of Graph Fourier MMD. This library includes the toy datasets which were used, including the grid graph (section 4), Minnesota graph, and Bunny graph. Source code for implementation of Graph Fourier MMD and the swiss roll experiment (section 4.1) is located on GitHub at https://anonymous.4open.science/r/Graph-Fourier-MMD-5C10/. For the biological experiments, cell-cell affinities were calculated using k-nearest neighbors upon centering and normalizing the data matrix of gene expression in each cell.

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

Grace X Y Zheng, Jessica M Terry, Phillip Belgrader, Paul Ryvkin, Zachary W Bent, Ryan Wilson, Solongo B Ziraldo, Tobias D Wheeler, Geoff P McDermott, Junjie Zhu, Mark T Gregory, Joe Shuga, Luz Montesclaros, Jason G Underwood, Donald A Masquelier, Stefanie Y Nishimura, Michael Schnall-Levin, Paul W Wyatt, Christopher M Hindson, Rajiv Bharadwaj, Alexander Wong, Kevin D Ness, Lan W Beppu, H Joachim Deeg, Christopher McFarland, Keith R Loeb, William J Valente, Nolan G Ericson, Emily A Stevens, Jerald P Radich, Tarjei S Mikkelsen, Benjamin J Hindson, and Jason H Bielas. Massively parallel digital transcriptional profiling of single cells. *Nat. Commun.*, 8(1):14049, January 2017.

## A  APPENDIX

### A.1  PROOF OF STATED RESULTS

#### A.1.1  PROOF OF THEOREM A.1.1

**Theorem** Let $C(\mathbf{L})$ be the column space of $\mathbf{L}$. Then $P - Q \notin C(\mathbf{L})$ if and only if there exists a connected component $C$ of $\mathcal{G}$ such that $\sum_{i \in C} P \neq \sum_{i \in C} Q$. Furthermore, if $d$ is a distance acting on $\mathcal{V}$ and $L$ is constructed so that $w(a, b) = 0$ when $d(a, b) = \infty$, then $W(P, Q) = \infty$ when $P - Q \notin C(\mathbf{L})$.

*Proof.* Note that if $C_1 \ldots C_m$ are the connected components of $\mathcal{G}$, then the kernel of $L$ is spanned by the mutually orthogonal vectors $\mathbb{I}\{C_1\} \ldots \mathbb{I}\{C_2\}$. First, if there exists a connected component $C_k$ such that $\sum_{a \in C_k} P(a) \neq \sum_{a \in C_k} Q(a)$, then,

$$\langle \mathbb{I}\{C_k\}, P - Q \rangle \neq 0$$

And we are done: $P - Q$ has nonzero component in the kernel of $L$. On the other hand, if $L$ has a nonzero projection in the kernel of $L$, this will be witnessed by some function in the basis. Thus, one

of our basis functions $\mathbb{I}\{C_k\}$ (for some $k$) will satisfy $\langle P - Q, \rangle \neq 0$, so looking at the corresponding connected component $C_k$,

$$\sum_{a \in C_k} P(a) \neq \sum_{a \in C_k} Q(a)$$

Giving us the other direction.

For the second part of the proof, assume $\mathcal{V}$ lie in some metric space with distance $d$ between vertices, and we define the Wasserstein metric $W(P, Q)$ by this distance. Also let $\Pi$ be the set of matrices $\pi$ which satisfy $\pi \mathbf{1} = P$, $\mathbf{1}^T \pi = Q$ and $D$ be the distance matrix defined by $D$ over $\mathcal{V}$. I now claim that, supposing $L$ is generated by an affinity $w$ such that $w(a, b) = 0$ if and only if $d(a, b) = \infty$, that if $P$ and $Q$ possess the above property, then $W(P, Q) = \infty$. Indeed, note that the set of transport plans is enclosed in a finite-dimensional polytope and is thus a compact set. So we may consider the transport plan $\gamma$. Suppose that $W(P, Q)$ is finite. Then for all $a, b$ such that $a$ and $b$ belong to different connected components of the graph, $\gamma(a, b) = 0$, or else $W(P, Q) \geq \gamma(a, b)d(a, b) = \infty$. Additionally, without loss of generality, there exists a $C_k$ for which $\sum_{a \in C_k} P(a) > \sum_{a \in C_k} Q(a)$. But since $\gamma$'s marginals are $P$ and $Q$, it should be that,

$$\sum_{a \in C_k} P(a) = \sum_{a \in C_k} \sum_{b \in \mathcal{V}} \gamma(a, b) = \sum_{a \in C_k} \sum_{b \in C_k} \gamma(a, b).$$

Likewise,

$$\sum_{a \in C_k} Q(a) = \sum_{a \in C_k} \sum_{b \in C_k} \gamma(a, b)$$

But $\sum_{a \in C_k} P(a) \neq \sum_{a \in C_k} Q(a)$, which contradicts our assumption that $W(P, Q) < \infty$. $\qquad\square$

### A.1.2 Proof of Corollary A.1.2

**Corollary** For a possibly disconnected graph $\mathcal{G}$ as defined,

$$\text{GFMMD}(P, Q) = \begin{cases} \sqrt{T}\|L^{-\frac{1}{2}}(P - Q)\| & \text{if } P - Q \in C(\mathbf{L}) \\ \infty & \text{otherwise.} \end{cases}$$

*Proof.* We repeat the same argument as before. First, if $P - Q$ is not in the column space of $L$, it has nonzero component in the kernel of $L$. Then, we conclude by Theorem A.1.1 that $P$ and $Q$ have different amounts of mass is a connected component $C_k$. Then let $f = \alpha \mathbb{I}\{C_k\}$. We then have that, $\mathbb{E}_P(\mathbb{I}\{C_k\}) - \mathbb{E}_Q(\mathbb{I}\{C_k\}) \neq 0$. So in particular, $\mathbb{E}_P(f) - \mathbb{E}_Q(f) = \alpha \cdot C$ for some nonzero number $C$. And since $\alpha \mathbb{I}\{C_k\}$ is in the kernel of $L$, $f^T L f = 0$. Therefore, $\text{GFMMD}(P, Q) \geq \alpha \cdot C$. Letting $\alpha \to \infty$, we know $\text{GFMMD}(P, Q) = \infty$. For the other direction, if $f \in C(\mathbf{L})$, then $\mathbf{L}\mathbf{L}^- f = f$, and the remainder of the proof is the same as that of Theorem 2. $\qquad\square$

### A.1.3 Proof of Theorem 5

**Theorem 5** If $X = \{\delta_i\}_{i \in \mathcal{V}}$ is a family of Kronecker-delta functions centered at each vertex of $\mathcal{G}$, then the $k$-dimensional embedding which best preserves the distances between signals in $X$ is equivalent up to rescaling to Hall's Spectral Graph Drawing of the Graph $\mathcal{G}$ in $k$-dimensions.

*Proof.* Note that $X$, the data matrix of Kronecker Deltas, is equal to $\mathbf{I}$, the $n$-dimensional identity. So $\sqrt{T}\mathbf{L}^{-\frac{1}{2}}X = \sqrt{T}\mathbf{L}^{-\frac{1}{2}}$, hence the best $k$-dimensional embedding of $\sqrt{T}\mathbf{L}^{-\frac{1}{2}}$ (respecting the $L^2$ norm between columns) will be equivalent to Principal Component Analysis (P.C.A.). Since $\mathbf{L}^{-\frac{1}{2}}\mathbf{1} = 0$, $\mathbf{L}^{-\frac{1}{2}}$'s columns are mean-centered, so its covariance matrix of $\sqrt{T}\mathbf{L}^{-\frac{1}{2}}$ is $\frac{T}{n}\mathbf{L}^{-\frac{1}{2}^T}\mathbf{L}^{-\frac{1}{2}} = \frac{T}{n}\mathbf{L}^-$.

Since its columns and rows are already mean centered. And thus P.C.A. will select the eigenvectors of $\mathbf{L}^-$ corresponding to the $k$th largest eigenvalues. Note that these are precisely given by $\psi_1, \psi_2..\psi_k$ with associated eigenvalues in $\mathbf{L}^-$ given by $\lambda_1^{-1} \ldots \lambda_k^{-1}$. Letting $\Lambda_k = \text{diag}(\lambda_1^{-1/2} \ldots \lambda_k^{-1/2})$ and $\Psi_k = (\psi_1 \ \ldots \psi_k)$, P.C.A. would embed $\sqrt{T}\mathbf{L}^{-\frac{1}{2}}$ as,

$$\Psi_k^T \mathbf{L}^{-\frac{1}{2}} = \Psi_k^T \Psi \Lambda^{-\frac{1}{2}} \Psi^T = (\mathbf{I}_k \quad \mathbf{0}_{n-k}) \Lambda^{-\frac{1}{2}} \Psi^T = (\mathbf{I}_k \Lambda_k \quad \mathbf{0}_{n-k}) \Psi^T = \Lambda_k \Psi_k^T.$$

So our embedding of distributions would be given by $\Lambda_k \Psi_k^T$. On the other hand, Hall's Spectral Graph Drawing would embed the graph $\mathcal{G}$ simply as $\Psi_k^T$, since it chooses the first $k$ nontrivial eigenvectors of $\mathbf{L}$. Thus, coordinates in each embedding are the same up to the rescaling by eigenvalues. $\qquad\square$

## A.2 GRID GRAPH

**Grid Graph** First, we consider a $16 \times 16$ grid graph (vertices given by $\{(i, j)\}_{1 \leq i, j \leq 16}$. We can construct a signal $P$ by placing a Dirac $\delta_{(8,4)}$ on the vertex (8,4) and then diffusing it with a heat filter (using time $\tau = 16$). $Q$ is generated likewise, but by applying a heat filter to $\delta_{(8,4+2j)}$ and diffusing for each $j = 0, 1, 2, 3$. The result are two modes: $P$ on the left, and $Q$ moving along the right. The distributions are visualized in the top row, and the witness function to their difference in the bottom row of figure 4.

And of course, the corresponding distances between $P$ and the $Q$'s (per the order presented above) are increasing in the distances between the appropriate centers.

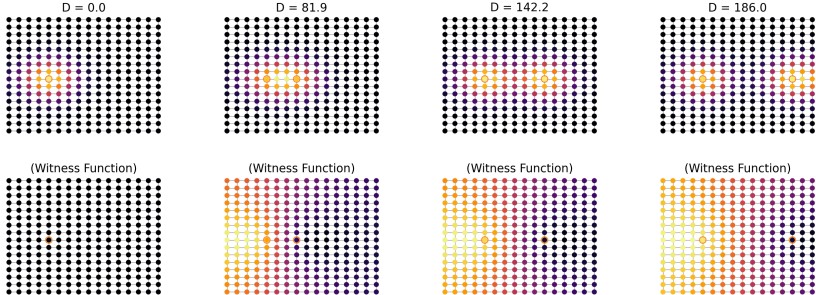

Figure 4: Top row: the distributions $P$ and $Q$, where the signal $P$ stays fixed but the vertex at which $Q$ is centered shifts to the right. Corresponding distances between distributions appear in the title, and the relevant centers of $P$ and $Q$ are highlighted. Bottom row: corresponding witness functions $f$ to the difference between $P$ and $Q$.

## A.3 DIFFUSING SIGNALS ON THE BUNNY GRAPH

One very simple sanity check of a measure of spread is to verify that the more we diffuse a Dirac, the lower the distance to the uniform. Indeed, if we begin with the Bunny graph (from pygsp's built in library) and diffuse the Dirac $\delta_{1400}$ (1400 was chosen for visual appeal) for scales $\tau = 2^0, 2^4, 2^8$, and $2^{12}$ (using a heat filter), we find that the corresponding measures of spread are 40.5, 26.9, 21.5, and 9.76. The signals are visualized below:

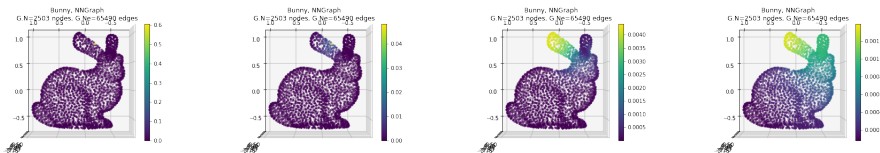

Figure 5: The signal $\delta_{1400}$ diffused to levels $1, 6^1, 6^2$, and $6^3$ using a heat filter.

### A.4 BIMODAL SIGNALS

We can take the earlier signals from the grid graph (each pair of $P$ and $Q$ for translations of $Q$) and combine them into a new signal $\frac{1}{2}(P+Q)$. This forms a family of bimodal signals for which the two modes spread. Accordingly, in the example above, the distance to the uniform is given by 11.14, 8.66, 6.13, and 6.09.

### A.5 LOCALIZATION ON THE MINNESOTA GRAPH

#### A.5.1 EXAMPLE: MINNESOTA GRAPH (BINARIZED)

A final sanity check for a measure of closeness to the uniform would be to begin with a density which puts all its mass on one vertex. Then, put equal mass on that vertex and its neighbors, then the neighbors of neighbors, etc. More specifically, let $N_k(i,j) = \{\exists k' \in [k] : A^{k'} > 0\}$, or $N_k(i,j) = \mathbf{1}\{\text{there is a path of length} \leq k \text{ from } i \text{ to } j\}$. Then we can consider multiplying this by a Dirac, say $\delta_0$ to get a family of signals. Using $k = 1, 4^1, 4^2, 4^3$, we have a family of distributions proportional to $N_1\delta_0, N_2\delta_0, N_3\delta_0$, and $N_4\delta_0$. Again, we can visualize the activated vertices in yellow:

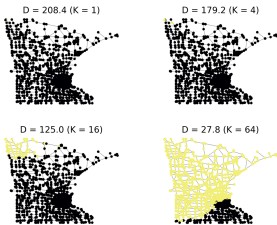

Figure 6: The zeroth vertex's neighbors, then neighbors of neighbors, etc. for order $1, 4, 16$, and $64$ neighbors. The corresponding distances to the uniform are given in the title.

### A.6 EXAMPLE: MINNESOTA GRAPH (SMOOTH WAVES)

A similar example we can consider is a similar class of signals which "spread" across the graph, but rather than activating neighbors, simply diffusing the signal from a given start vertex. Here, we choose the same start vertex, and run heat diffusion at times $\tau = 2^0, 2^4, 2^8$, and $2^{12}$.

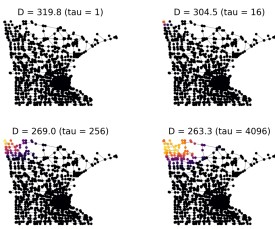

Figure 7: Visualization of the diffusions of the signal $\delta_0$ at times $2^0, 2^4, 2^8$, and $2^{12}$. The corresponding distances to the uniform are given above.

