# OpenReview forum: "Graph Fourier MMD for signals on data graphs"
_ICLR.cc/2023/Conference — Submitted to ICLR 2023_

### Official Review · Reviewer_yZp1 · 2022-10-23

**Confidence:** 4
**Clarity, Quality, Novelty And Reproducibility:** 1.  Could the authors elaborate the a…
**Correctness:** 4
**Technical Novelty And Significance:** 3
**Empirical Novelty And Significance:** 3
**Recommendation:** 6

**Strength And Weaknesses:**

Strength

1. The origin of the question has been clearly described.

2. Propose a new distance as specified in Definition 4.

3. The theoretical analysis and derivation of solution are solid.

Weakness

1. The experimental examples are not very significant. This reflects the fact or question how this theory can benefit in wide learning scenarios.

**Summary Of The Paper:**

This paper proposes a way to define a metric (called GFMMD) for non-negative graph signals in a way to account for geometric structure. It comes from by mimicing the so-called Integral Probability Metrics, particularly 1-Wasserstein distance for discrete distributions. GFMMD can be computed rapidly (relying on approximation to L^{-1/2}) and in closed-form.  From this closed-form solution, an embedding can be defined.  This newly defined distance has been applied in several empirical examples. It seems certain benefits can achieve.

**Summary Of The Review:**

The idea in the paper is simple but interesting. It emphasizes the importance of L^{-1/2}f although the paper did not go further in this direction. Overall the paper is well presented. More interesting applications are desired.

---

### Official Review · Reviewer_tGJZ · 2022-10-24

**Confidence:** 4
**Correctness:** 2
**Technical Novelty And Significance:** 2
**Empirical Novelty And Significance:** 2
**Recommendation:** 3

**Clarity, Quality, Novelty And Reproducibility:**

The article is too unclear and quite incremental. There is a lot of missing references and important competing methods that are not mentioned

**Strength And Weaknesses:**

The idea of this distance is simple and seems quite natural. The advantage is that it is reasonably simple to calculate. However, my overall impression of the article is that it doesn't seem mature enough to be published yet, and many points are unclear.

**Summary Of The Paper:**

This article introduces a distance between probability distributions on a graph. It is defined as the distance in the RKHS of the embedding of these distributions according to a kernel defined by the pseudo-inverse of the Laplacian of the graph.

**Summary Of The Review:**

Here are my detailed comments:

- Litterature review and competing methods:

My main reproach concerning this article is that it is not complete enough and that the literature review on the subject is too light. The article lacks comparisons with many existing methods which are very similar to the proposed method. In this context the sentence "relatively little attention has been given to computing such distances for distributions on graphs" is very misleading.

On the one hand, all the literature on spectral embedding methods, or more generally on harmonic analysis on graphs, is almost not mentioned in the article. However, this is exactly what is proposed by the authors to compare signals on graphs seen as probability distributions. Indeed Theorem 2. simply tells us that the chosen metric corresponds exactly to computing the Euclidean distance between the spectral embeddings of the signals. From a shape point of view, many works have already addressed the issue (e.g. [1, 2]) and others have tried to generalize these approaches for the case of signals on different graphs e.g. [3]. In this context, lernel embedding of signals based on the Laplacian is far from being new. From an OT persepective there is also a lot of litterature on the exact same problem that is not mentionned e.g. [4-6] (even for signals on different graphs e.g. [7-10]). In light of these various works, the proposed method is quite incremental. I think it is necessary to position the approach well with respect to the spectral and other transport variants that already exist, without omitting them, and to compare with them.

- Justification of the approach:

I also find that talking about Wasserstein distance to define the GFMMD metric is rather clumsy, even a bit forced. Indeed the authors artificially start from the distance of $W_1$ and relax the constraints, going from a Lipschitz constraint to a constraint on the sum. The justification given is "This condition is more robust to noise than the corresponding 1-Lipschitz constraint, and results in a closed-form solution". The second sentence is indeed correct but the first one lacks justification. Moreover, to define GFMMD there is no need for the Wasserstein distance. It is enough to define the discrete kernel $K = L^{-}$ and to consider the MMD associated to this kernel (as illustrated in Theorem 6.). Going through the Wasserstein distance instead of presenting the simple definition of GFMMD with the explicit kernel reinforces the unclear side of the paper.

More importantly, there is a real confusion between "signals on graphs" and "probability distributions on graphs". Indeed, the way it is written, $GFMMD(P,Q) = \|L^{-\frac{1}{2}}(P-Q)\|$ implies that the method cannot take into account signals on the graph. Indeed "$P$ is a probability distribution on the vertices/nodes": thus it is simply a histogram (say the graph have two nodes $1, 2$ then $P$ is for example $(1/2, 1/2)$). So $\|L^{-\frac{1}{2}}(P-Q)\|_{2}$ is simply the $\ell_2$ distance between the histograms reweighted by the Laplacian, and there is no signal here: unless $P$ is defined as $(f(1), f(2), \cdots , f(n))$ where $f(i) \in \mathbb{R}$ is the value of the signal at the node $i$. But in this case $P$ is not a probability distribution anymore since $f(i)$ can be negative: it is a signed measure. This confusion is also supported by the next point.

- About "Empirical Results \& Applications":

The first paragraph of this part is indeed quite surprising: I have the impression that the authors confuse "negative signals $P$ and $Q$" with "negative measures/distributions $P$ and $Q$". In the first case there is no problem to consider the MMD/Wass between two probability distributions associated to signals which can take negative values (one can easily compute the MMD or Wass distance between $\frac{1}{2} \delta_{-1} + \frac{1}{2} \delta_{-2}$ and $\frac{1}{2} \delta_{1} + \frac{1}{2} \delta_{12}$). In this case, why mention the fact that we have to normalize? In the second case, $P$ and $Q$ are indeed no longer probability distributions and the framework/formalism must be changed.

I have trouble seeing the point of the grid graph experiment: what is exaclty this "witness function"? The distance $D$ is, I guess the GFMMD, right? What this example shows is that the GFMMD increases when the $2$ signal moves away from the $1$ signal on the graph, right?

I am not really convinced by the experiment 4.1 which tries to show that GFMMD is a good measure of similarity between distributions on the graph. The example is quite artificial and the results are really noisy, it is hard to understand why one method is really better than the others (for example Diffusion EMD vs GFMMD)

- Other comments:

Overall I find that the Figure 1, 2 and 5 are really hard to read. I think it is important to make the titles and figures larger.

The article about the package "python optimal transport" should be quoted instead of just mentioning it in a sentence. Similarly references Chebyshev polynomials applied to filter approximation on graphs are missing (e.g. [11]).

The "Hall's Spectral Graph Drawing of the Graph G in k-dimensions" is not properly defined in the article and there are no references, so we don't really understand what Theorem 7 is about.

I think the writing of the article could be improved. In particular, the first three pages contain a lot of definitions that I think are not really useful and could be removed to discuss many more important points like the links with the different spectral approaches or to discuss other transport approaches for signals on graphs.

Refs:
[1] Michael M. Bronstein, Alexander M. Bronstein. Shape recognition with spectral distances. IEEE TRANS. PAMI 2010.

[2] Rustamov, Raif M. Laplace-Beltrami Eigenfunctions for Deformation Invariant Shape Representation. Proceedings of the Fifth Eurographics Symposium on Geometry Processing. 2007

[3] Saurabh Verma, Zhi-Li Zhang. Hunt For The Unique, Stable, Sparse And Fast Feature Learning On Graphs. NeurIPS 2017.

[4] Montacer Essid, Justin Solomon. Quadratically-Regularized Optimal Transport on Graphs. SIAM J. SCI. COMPUT. 2018

[5] Tam Le, Truyen Nguyen, Dinh Phung, Viet Anh Nguyen. Sobolev Transport: A Scalable Metric for Probability Measures with Graph Metrics. AISTATS 2022.

[6] Le, T., Yamada, M., Fukumizu, K., and Cuturi, M. Tree-sliced variants of Wasserstein distances. NeurIPS 2019

[7] Nikolentzos, G., Meladianos, P., and Vazirgiannis, M. Matching node embeddings for graph similarity. In Proceedings of the Thirty-First AAAI Conference on Artificial
Intelligence, February 4-9, 2017, San Francisco, California, USA., pp. 2429–2435, 2017.

[8] Titouan Vayer, Laetitia Chapel, Rémi Flamary, Romain Tavenard, Nicolas Courty. Optimal Transport for structured data with application on graphs. ICML 2019

[9] A. Barbe, M. Sebban, P. Gon¸calves, P. Borgnat and R. Gribonval. Graph Diffusion Wasserstein Distances. ECML 2020.

[10] Hongteng Xu, Dixin Luo, Hongyuan Zha et Lawrence Carin Duke. Gromov-Wasserstein Learning for Graph Matching and Node Embedding. ICML 2019

[11] D. I. Shuman, S. K. Narang, P. Frossard, A. Ortega, and P. Vandergheynst. The emerging field of signal
processing on graphs: Extending high-dimensional data analysis to networks and other irregular domains.
IEEE Signal Processing Magazine, 30(3):83–98, 2013.

---

### Official Review · Reviewer_SH8k · 2022-10-24

**Confidence:** 4
**Correctness:** 3
**Technical Novelty And Significance:** 2
**Empirical Novelty And Significance:** 3
**Recommendation:** 5

**Clarity, Quality, Novelty And Reproducibility:**

The article has some novelty, even if I question the real novelty: what is everything derives from the properies of the kNN  graph ?
If so, what would be the impact of changing that, or of having a structure not well approximated by this k-NN graph.

For Reproducibility, a code is provided.

The article is globally clear and of good qualoity. Some weak aspects can be revised, as states above.

**Strength And Weaknesses:**

Strength

- The proposed method, as summarized by Definition 4 and Theorem 2, is an interesting approach to design a distance between two multivariate distributions.

- there are some interesting numerical experiments in Section 5 that provide a correct grasp of the potentialities of the work.

- the article is mostly clear and well written.

Weaknesses

1- Some points shoud be better stated: the initial result in 3.1 assumes a distance function $d(a,b)$ between all pairs of nodes yet the author suggest just before Def 4 and again afterwards to use instead an affinity graph between nodes instead of $1/d^2$. Is this affinity graph truncated or sparsified) in some way ? Or is it intended to gave a weighted full graph ?

2- Also: Algorithm 1 takes $X$ as an input and never is considered the more general (and interesting) situation where $X$  and $\mathcal{G}$ which is not a k-NN graphs over $X$. Would the method be amenable to such a more general situation ?
And, if not, is the performance of the method only a conequence of the approximation of some manifold where the data points lie thanks to the k-NN graph ? Could topologies more varied than ones derived from data on swiss roll be considered ?

3- Can you challenge my impression that, in 4.1, the works merely rely on the possibility to approximate the structure (manifolds) thanks to k-NN ? If that so, what is actually the difference as compared to Laplacian embedding methods ?

4- I don't understand why most of A.1 is not in the main text, while discussion about disconnected graphs (half or page 5, including Theorem 5 and the corollary) could safely be postponed to appendix, as there is almost no practical usage of working on disconnected graphs without preprocessing by connected components.

5- Some references are missing: to Hall's spectral drawing ; to references about Chebyshev polynomial approximation for graph filters ; in 4.1 when listing the various methods for comparisons

Notes / questions:

- MMD should be "translated" in the introduction

- I am not confident that the lifting approach described at the beginning of 4 will not have major impact. Has this point been studied more ?

- for the grid graph and Fig 1, how are chosen the diffusion times ?

- The aspects about Gene Locality is not clear for me (but maybe it's because it's a field that I know less).


**Summary Of The Paper:**

The article defines a measure of Maximum Mean Discrepancy (MMD) for Graph Fourier features. it discusses how one could use the Laplacian of a graph suited to multivariate data so as to compute a GF-MMD. Some empirical results and applications are shown in an extensive numerical section.

**Summary Of The Review:**

My appreciation is borderline for acceptation; I put currently marginally under the acceptance threshold.

The weaknesses in the presentation and on some insights should be corrected by the authors.

---

### Official Review · Reviewer_3iNi · 2022-10-26

**Confidence:** 4
**Correctness:** 3
**Technical Novelty And Significance:** 2
**Empirical Novelty And Significance:** 2
**Recommendation:** 3

**Clarity, Quality, Novelty And Reproducibility:**

The authors propose a new distance, Graph Fourier MMD (GFMMD) for distributions on graphs. However, its motivation and advantages over existing distances in the literature are not clear.

I have some following concerns:
+ What is the advantage of the proposed distance (GFMMD) over recent proposed distance for distributions on a graph: e.g., Diffusion EMD, Sobolev transport

Ref:
Tong, A.Y., Huguet, G., Natik, A., MacDonald, K., Kuchroo, M., Coifman, R., Wolf, G. and Krishnaswamy, S., 2021, July. Diffusion earth mover’s distance and distribution embeddings. In International Conference on Machine Learning (pp. 10336-10346). PMLR.

Le, T., Nguyen, T., Phung, D. and Nguyen, V.A., 2022, May. Sobolev Transport: A Scalable Metric for Probability Measures with Graph Metrics. In International Conference on Artificial Intelligence and Statistics (pp. 9844-9868). PMLR.

+ For the formulation of GFMMD in Theorem 2, it is a Mahalanobis distance (with matrix L instead of identity matrix as in L2 distance) over the full graph. When input distributions P, Q are very sparse on graph, the proposed GFMMD is still computed on the size of the entire graph, it may affect its advantage on the large-scale graph.

+ In Section 2.1., why the author call graph G a distance graph when the edges are computed by affinity (similarity between corresponding two nodes). Is there any difference about the meaning when one use (i) kernel or (ii) distance to build the graph?

+ For arbitrary graph, the weight on the graph may be negative, it is unclear why the identity of f^TLf (in Section 2.1) can imply the positive semidefiniteness for L

+ Could the author elaborate the relaxed constraint which is more robust to noise? (In section 3.1)

+ In the proof of Lemma 4, GFMMD is positive definite? (or negative definite?) or simply nonnegative?

+ In Algorithm 1, the role of Chebyshev polynomial approximation is important to reduce the complexity of the proposed method. It is better if the authors elaborate it with more details and discuss the trade-off about the quality of approximation with the gain from computation.

+ In Algorithm 1, what is the complexity to build the kNN graph over X with O(nlogn) edges? How does one to choose K to guarantee that we obtain a graph with O(nlogn) edges? Is the built graph connected?

+ For experiment 4.1, the authors should elaborate what is the ground truth (exact EMD)? what is its cost metric? Why the correlation with the Exact EMD is important over several baselines? I am confused that why the proposed Graph Fourier MMD is more correlated with Exact EMD? (as far as I understand MMD and EMD are two different instances of IPM!)

+ For experiment 4.2, the authors should compare the proposed approach with other baseline distances for distributions on a graph, besides the Wilcoxon rank sum differential expression.
------
After the rebuttal:
I thank the authors for the rebuttal. The submission necessarily needs a major revision and one more round for review is required.
The rebuttal has clarified some raised points. However, I respectfully disagree some other points (especially, a disadvantage of computation over sparse input distributions but it requires to consider the entire graph -- although the closed form for computation is a plus, but the computation involves the whole graph (manifold) for sparse input distributions is clearly not a good point; "positive" and "positive definite" for a distance are two different notions; it is also better to clearly describe how to build a (connected) graph with certain number of edges with its complexity analysis, etc...)







**Strength And Weaknesses:**

Strength:
+ The authors prove a new distance, namely Graph Fourier MMD for distributions on graphs

Weaknesses:
+ It is not clear the advantages of the proposed distance over existing ones in the literatures for distributions on a graph.
+ The empirical advantages of the proposed method are unclear.  (e.g., no baselines about distances for distributions on graphs on the single cell analysis?, it is unclear why it is important to measure the correlation with the exact EMD?)


**Summary Of The Paper:**

The authors propose a new distance between distributions (or non-negative signals) on graphs, namely Graph Fourier MMD. The authors also prove its scale invariance and applicability to disconnected graphs. The authors evaluate the proposed distance for gene clusters.

**Summary Of The Review:**

The proposed method has some potential. However, it is better in case the authors elaborate the proposed approaches with more details and improve the experiments.

---

### Decision · Program_Chairs · 2023-01-20

**Decision:**

Reject

**Justification For Why Not Higher Score:**

More detailed discussion comparing the results with respect to prior work is necessary.

**Justification For Why Not Lower Score:**

N/A

**Metareview: Summary, Strengths And Weaknesses:**

The paper proposes a distance to measure signals on graph structured data. It adapts the Maximum Mean Discrepancy to arrive at
a new measure with GFMMD) and provably shows it is an instance of IPM.  It  further uses Chebychev polynomials to compute the function. There is wide appreciation of the contributions. However the main concern seems to be that positioning of the paper
vis-a-vis past work is not clear. It would be useful if some of the methods raised by the referees are comprehensively discussed to accentuate their difference with existing works. This will help in ensuring better appreciation of the contributions.

**Summary Of Ac-Reviewer Meeting:**

Not applicable